# Kinome-Wide Virtual Screening by Multi-Task Deep Learning

**DOI:** 10.3390/ijms25052538

**Published:** 2024-02-22

**Authors:** Jiaming Hu, Bryce K. Allen, Vasileios Stathias, Nagi G. Ayad, Stephan C. Schürer

**Affiliations:** 1Dr. John T. Macdonald Foundation Department of Human Genetics and John P. Hussman Institute for Human Genomics, Miller School of Medicine, University of Miami, Miami, FL 33136, USA; jxh1011@miami.edu; 2Department of Molecular and Cellular Pharmacology, Miller School of Medicine, University of Miami, Miami, FL 33136, USA; allenbk4@gmail.com (B.K.A.); v.stathias@med.miami.edu (V.S.); 3Institute for Data Science & Computing, University of Miami, Miami, FL 33136, USA; 4Center for Therapeutic Innovation Miller School of Medicine, University of Miami, Miami, FL 33136, USA; na853@georgetown.edu; 5Miami Project to Cure Paralysis, Department of Psychiatry and Behavioral Sciences, Miller School of Medicine, University of Miami, Miami, FL 33136, USA; 6Sylvester Comprehensive Cancer Center, Miller School of Medicine, University of Miami, Miami, FL 33136, USA

**Keywords:** virtual screening, kinase drug discovery, computational kinase profiling, machine learning, multi-task deep learning

## Abstract

Deep learning is a machine learning technique to model high-level abstractions in data by utilizing a graph composed of multiple processing layers that experience various linear and non-linear transformations. This technique has been shown to perform well for applications in drug discovery, utilizing structural features of small molecules to predict activity. Here, we report a large-scale study to predict the activity of small molecules across the human kinome—a major family of drug targets, particularly in anti-cancer agents. While small-molecule kinase inhibitors exhibit impressive clinical efficacy in several different diseases, resistance often arises through adaptive kinome reprogramming or subpopulation diversity. Polypharmacology and combination therapies offer potential therapeutic strategies for patients with resistant diseases. Their development would benefit from a more comprehensive and dense knowledge of small-molecule inhibition across the human kinome. Leveraging over 650,000 bioactivity annotations for more than 300,000 small molecules, we evaluated multiple machine learning methods to predict the small-molecule inhibition of 342 kinases across the human kinome. Our results demonstrated that multi-task deep neural networks outperformed classical single-task methods, offering the potential for conducting large-scale virtual screening, predicting activity profiles, and bridging the gaps in the available data.

## 1. Introduction

Precision therapy attempts to optimize the treatment of diseases for specific patient sub-populations, defined by clinical biomarkers [1]. It is especially promising in the context of cancer therapy, where patient responses to therapy can vary considerably and often lack clinical durability, even with the advent of single-targeted therapies that address specific tumor genomic amplifications and disrupt oncogenic signaling processes [2]. While patients often respond to targeted therapy initially, resistance is commonly detected as heterogeneity in the tumor cell population, which allows for the adaptive selection of sub-populations that are not affected by the drug, or cells effectively rewire their signaling machinery to bypass the drug inhibition [3]. This paradigm has facilitated the acceptance of targeted polypharmacology and combination therapy strategies to address drug resistance and reduce recurrence in cancer patients [4].

The success of small-molecule kinase inhibitors in cancer therapy is evident, with over 70 FDA-approved drugs in this class [5,6]. While receptor tyrosine- and cyclin-dependent kinases have been established as target groups, it is increasingly recognized that a much larger proportion of the human kinome is likely therapeutically relevant [7]. Beyond cancer, kinase drug discovery programs now span a wide range of targets and disease areas [8]. Despite their efficacy, kinase inhibitor drugs often exhibit limited response rates, and their effectiveness is of a short duration, necessitating the need for precision therapeutic approaches as we learn from the integration of clinical data, such as drug responses and cancer multi-omics characterization, and which features of these molecules are likely to provide the best outcome [9,10,11].

In silico methods, such as ligand-based virtual screening, have been applied to kinase activity modeling and have shown that the utilization of known kinase small-molecule topological and bioactivity information can lead to the enrichment of novel kinase active compounds [12]. The advantage of ligand-based virtual screening is that compounds can be readily evaluated for interactions across all kinases for which applicable bioactivity data are available. If such models were sufficiently accurate, they could support the prioritization of compounds with a desirable polypharmacology profile while deprioritizing those compounds that may inhibit kinases that are therapeutically less relevant or lead to toxicity in a specific disease or patient [13,14].

Classification approaches for ligand-based virtual screening, also called target prediction or target fishing, include a variety of single-task machine learning algorithms, such as logistic regression, random forests, support vector machines, and naïve Bayesian classifiers that aim to separate each kinase target individually, whether a compound belongs in the active or inactive class [15,16,17]. Although these single-task methods perform relatively well in many instances, they do not take into consideration the membership of molecules in multiple classes and therefore do not adaptively learn across different categories, limiting their applicability to predict profiles. To achieve a better performance, machine learning methods must combine diverse sources of bioactivity data across multiple targets [18]. This is especially relevant for kinases, where there is a large degree of similarity across many different kinases and their inhibitors [19]. 

Motivated by previous advancements in multi-task neural network architectures, we conducted an evaluation of multi-task deep neural networks (MTDNN) for kinome-wide small-molecule activity prediction, comparing their performance to single-task methods [20]. We compiled an extensive dataset comprising over 650,000 aggregated bioactivity annotations for more than 300,000 unique small molecules, covering 342 kinase targets. We evaluated the machine learning methods using reported actives and reported inactives as well as using the reported actives and considering all other compounds in the global dataset as inactives. Our results indicate that multi-task deep learning results in a substantially better predictive performance compared to single-task machine learning methods using various cross-validation strategies. Additionally, the performance of the multi-task method continues to improve with the addition of more data, whereas single-task methods tend to plateau or diminish in performance. Our work extends the contribution of prior studies, which employed deep learning techniques for broad target classification and specific benchmarking datasets [19,21,22,23,24,25,26,27], by evaluating MTDNN on a much larger dataset than previously explored, utilizing real-world datasets from both public and private sources. Through thorough characterization and evaluation, our study provides practical insights into the applicability of kinome-wide multi-task activity predictors for virtual profiling and compound prioritization.

## 2. Results

To investigate the implications and applicability of multi-task deep learning to kinome-wide small-molecule activity prediction, we addressed the following research questions:Can multi-task neural network architectures enhance the predictive performance of kinase classification compared to single-task methods and non-deep learning multi-task approaches?Compared to single-task methods, how effectively do multi-task models generalize, and what is the impact of data availability on their performance? Do these models exhibit an improved performance with larger data or a diminished performance with smaller data points?Are multi-task neural network architectures applicable to account for domain-specific differences between kinase groups? In other words, do they perform across the kinome despite large variations in available data?Can the multi-task models be utilized to prioritize small molecules with kinase polypharmacology in new datasets, or are they limited to the specific dataset they were trained on?

### 2.1. Predictive Performance Comparison

To address our first question, we built and evaluated single- and multi-task classifiers using aggregated kinase bioactivity data obtained from ChEMBL [28] and KKB [27]. Together, our dataset comprised 668,920 activity data points, covering 342 targets across the human kinome and featuring 315,604 unique compounds (Figure 1 and Figure 2 and Appendix A). The known active–known inactive (KA–KI) models were evaluated first, which employed only the reported active and inactive compounds for each task. To address the issue of limited samples when using only the KA–KI set to train the model, we created the known active–presumed inactive (KA–PI) set based on reported actives and considering all other compounds in the set as inactive to fulfill the purpose of virtual screening by leveraging the entire dataset. Given that deep learning approaches tend to perform better with larger datasets, we expected that the model performance would be better to retrieve a small subset of actives from a larger dataset using the KA–PI assumption. This approach is also a more realistic scenario for virtual screening where the vast majority of compounds are inactive. 

The receiver operating characteristic (ROC) curve is a widely used performance metric in machine learning, particularly for binary classifications. It shows the relationship between the true positive rate and the false positive rate at different classification thresholds. A higher area under the ROC curve, referred to as the ROC score, indicates a better model performance in distinguishing between positive and negative samples. Figure 3 and Figure 4 provide visual representations of the ROC score distribution and the impact of active compounds on the ROC score, respectively, for each machine learning method applied to all kinase tasks. Both the MTDNN model and single-task methods were evaluated using the random stratified splitter, as outlined in Section 4. For the MTDNN model, a random cross-validation strategy was employed, and it was compared to traditional single-task machine learning methods in both figures. 

As expected, the KA–PI models performed significantly better than the KA–KI models across all machine learning methods based on ROC scores. Figure 3 and Figure 4 clearly illustrate that the KA–PI MTDNN model achieved the highest ROC score among all the machine learning methods used, outperforming the performance of the single-task models, and the KA–PI models performed especially well for tasks containing large amounts of active compounds compared with the KA–KI models. A statistical analysis was conducted using a Mann–Whitney U test to compare the performance of MTDNN vs classical single-task learning methods based on the ROC score metric (Table 1).

KA–PI MTDNN’s performance was compared further with single-task deep neural networks (STDNNs) to provide a thorough assessment of the model performance. We found that the MTDNN generally outperformed the STDNNs (Appendix A). Particularly, MTDNN performs better than the STDNNs for tasks with lower data points, especially for kinases in the TK and AGC groups (Appendix A). This difference in performance is likely due to the inherent differences between multi-task and single-task learning, where the multi-task model leverages shared knowledge and representations across various tasks, leading to an enhanced overall performance compared to the more specialized single-task learning approach. 

Additionally, to discern whether the enhancement in the model performance resulted from the deep learning algorithm or the multi-task learning approach, we conducted a comparison between the KA–PI MTDNN model and a multi-task random forest (MTRF) model (Appendix A). While previous comparisons demonstrated the superiority of multi-task over single-task learning, this comparison with MTRF points out that these results also attribute the performance improvement to the deep learning algorithm rather than solely to the multi-task approach.

### 2.2. Data Efficiency and Generalization of Multi-Task Models

To investigate our second research question, which focuses on the generalization capabilities of the multi-task neural network models, we assessed their performance on tasks with varying numbers of active compounds. Interestingly, compared to single-task methods, the multi-task neural network predictions did not degrade as rapidly on tasks with smaller numbers of actives and continued to perform well even for tasks with less than 50 active compounds (Figure 4). Enrichment of recovered actives out of a set of mostly inactives is the most relevant performance indicator in virtual screening. The KA–PI models are most relevant in that context. When comparing the model performance based on the normalized enrichment factor (EF/EFmax) at 0.1% of the tested compounds across all kinase tasks, the multi-task deep learning approach consistently outperformed single-task classification methods (Figure 5). This superiority is likely due to the utilization of shared hidden representations among various kinase prediction tasks. By sharing the weights of hidden layers across tasks, only the output weights are task specific, which allows for the network to leverage commonalities between different tasks and to achieve better generalization with reduced overfitting. Our hyperparameter optimization strategy revealed that networks with two hidden layers, consisting of 2000 and 500 neurons, resulted in the best performance across all evaluated metrics for all kinase tasks. Networks with three or more hidden layers did not yield any major improvements nor did they decrease in the predictive performance. However, it is worth noting that training networks with a larger batch size did result in a decrease in performance. While a larger batch size can reduce the model training time, this increase in efficiency is not justified, as the weighting parameters are not able to adequately fine-tune themselves through each epoch.

To further evaluate the KA–PI models, which we considered more generally applicable, we evaluated the model performance as a function of the number of active compounds based on the ROC score from a five-fold cross-validation (Figure 6). Multi-task deep learning consistently outperformed the single-task classification methods. Indeed, the MTDNN model continued to perform well even for tasks with fewer active compounds. Appendix A illustrates that enrichment decreased for single-task modeling methods for kinase tasks with fewer than 100 active compounds, while the MTDNN model was less affected. Moreover, for classes with less than 50 active compounds, the multi-task method achieved close to the maximum enrichment (1000 for kinase tasks with less than 0.1% actives), surpassing the performance of the single-task methods. We validated that the KA–PI MTDNN model indeed learns the small-molecule activity data as a function of topological indices of the chemical structure by randomizing the kinase activity labels while maintaining the number of actives in each dataset. As expected, this validation process resulted in 5-fold cross-validation ROC scores of approximately 0.5. This corresponds to random classification, indicating the modeling approach did not overfit the data and reinforcing the applicability of the KA–PI MTDNN method.

To explore how well MTDNN models generalize compared to the single-task methods, we investigated the impact of different cross-validation strategies. Specifically, we split the data based on the chemical scaffold, by molecular weight, and randomly. We evaluated different strategies for splitting compounds by scaffold, including Murcko scaffolds and clustering based on topological descriptors. We found that clustering our dataset into approximately 300 clusters with an average of around 1000 compounds each resulted in chemically distinct clusters (Appendix A), which worked better than Murcko fragments. This was mainly due to the large number of Murcko scaffolds in our dataset (>100,000), which resulted in training and test sets that resembled random splitting. Across all cross-validation strategies, which involved splitting the dataset based on chemical scaffold/cluster, molecular weight, and random order, we trained the model five times, each time training on four subsets, and evaluated the held-out subset. Hyperparameter optimization was employed for all models, and a 10% validation split was used for validation and testing purposes. The hyperparameters exhibited stability across different data splits, ensuring a consistent and reliable performance. As expected, ROC scores were higher for models cross-validated by random splitting and worse for models evaluated with a different scaffold (Appendix A). As observed before, the single-task model performance decreased significantly for kinases with fewer active compounds in contrast to the MTDNN approach, which continued to correctly classify active compounds even for those kinases that have a low number of actives.

In addition to ROC scores and arguably more relevant to virtual screening, we evaluated the different cross-validation splitting strategies based on enrichment; specifically, we evaluated how many true positives are recovered in 0.1% of tested samples depending on the ratio of actives in a dataset (Figure 7). As the number of active compounds in the test set increased, the ability to correctly rank the active compounds within the top 0.1% of the test set improved. Therefore, as anticipated, the fraction of true positives in a 0.1% test set on average increased with the ratio of actives to total compounds and reached close to 100% as the latter reached above 0.1%. Considering the maximum possible enrichment as the inverse of the ratio of actives to total compounds, many of the models achieved a nearly complete recovery of all actives in the test set. In practical terms, if a selection of 1000 compounds was made from a pool of one million, the best models would have identified between 10 and close to 1000 actives. However, it should be noted that the best models assume the training set to be representative of the test set. Cross-validation after splitting by scaffold or molecular weight performs worse than random splitting. Nevertheless, even for new scaffolds, the MTDNN models exhibited potential applicability for virtual screening. 

In addition to the different cross-validation splitting strategies, we also explored how the model performance would scale with the available data by generating an additional dataset that contained the same number of total compounds but only kept class labels exclusively for the top 100 kinase classes with the most active molecules. To assess data dependence, we randomly selected 10%, 20%, 30%, 50%, and 100% of that data. An evaluation of models across these different data thresholds revealed a consistent pattern. The multi-task method exhibited sustained increases in enrichment as the amount of available data increased, indicating its ability to leverage a larger dataset for improved performance (Appendix A). Conversely, the performance of single-task methods displayed more gradual and softer increases, eventually plateauing or even decreasing in performance. These findings highlight the superiority of the multi-task approach in utilizing available data effectively, resulting in enhanced performance across a range of data thresholds.

### 2.3. Multi-Task Neural Networks for Kinase Profiling across the Kinome

To gain deeper insights into the performance of multi-task and single-task machine learning models for kinome-wide small-molecule prediction, we examined the impact of dataset similarities on the model performance (Figure 8). All active compounds for each kinase task were compared globally to all other kinase task actives, and the average maximum chemical similarities were calculated (see Section 4. Figure 8 illustrates the relationship between the ROC score and chemical similarity to other kinase tasks. Notably, the multi-task DNN, in contrast to the single methods, performs particularly well for kinase tasks that share similar compounds with other kinase datasets. 

We also investigated the model performance across kinase target protein families, defined by kinase groups in the Drug Target Ontology (DTO) [29]. Each kinase task was organized into its corresponding group based on the DTO (see Section 4), and model evaluation metrics were visualized accordingly to illustrate differences in predictive performance. We observed that the multi-task models maintained high predictive performance across and within all kinase groups, even for those tasks that do not contain many active examples and are underrepresented globally when compared to single-task methods (Appendix A). In contrast, single-task methods exhibited a higher degree of variability in their results across and within each kinase group task. While they performed well for kinases with a substantial number of active compounds, their performance decreased for groups with fewer actives. On average, single-task methods showed better performance for groups that have many kinase tasks with a large number of actives. However, these methods were unable to effectively leverage information from even the most similar kinases, as they lacked the ability to cross-leverage data from other tasks. 

### 2.4. Applicability of the Multi-Task Model on External Datasets

To assess the general applicability of the MTDNN model to novel experimental data, we evaluated the performance of the KA–PI models in predicting the activity of compounds used in the LINCS KINOMEscan profiling data available from the LINCS Data Portal [30,31] (see Section 4 for details). The multi-task deep learning model achieved an accuracy of approximately 85% in classifying compound activity across the various kinases. This accuracy level indicates that the majority of the predicted actives are indeed true actives, suggesting that the model can predict external independent compound activity across a diverse set of kinases with a reasonably high level of precision (Appendix A). Additionally, a spreadsheet and an upset plot were included in the supporting document (Appendix A), which contains a list of true active compounds sourced from LINCS KINOMEscan profiling data, together with their SM_LINCS_ID obtained from the LINCS Data Portal (LDP), highlighting potential candidates that exhibit activity against multiple kinase tasks.

We further expanded our investigation to include the IDG-DREAM round 1 and round 2 kinase inhibitor datasets, which were obtained from the IDG-DREAM Challenge, a collaborative effort aimed at mapping the unexplored target space of kinase inhibitors [32] (see Section 4). By including these datasets, we aimed to evaluate the performance of our MTDNN model on a diverse range of novel kinase inhibitors, thereby broadening the scope of our analysis of model applicability. The model performance using IDG datasets yielded a mean ROC score of 0.78, which can likely be attributed to differences in the kinase inhibitor chemical structures but also to the nature of the models intended for virtual screening.

As illustrated in Figure 8, the average maximum similarity between compounds from different kinase classes tended to be above or around 0.8. In contrast, Tanimoto similarities between the training data and the IDG dataset compounds were mostly below 0.5, with only a small fraction (less than 10) exhibiting scores above 0.8. The IDG datasets consisted of primarily actives (63%, according to our definition outlined in the Section 4), leading to less favorable ROC statistics compared to our model. Looking at the model precision, we found ~70% of the predicted actives were actual actives. 

## 3. Discussion

In this study, we investigated the MTDNN approach for one of the largest protein families of drug targets, kinases, which are particularly relevant in cancer. We leveraged large numbers of available public and private small-molecule activity data, modeled the data in different ways, and compared deep learning and multi-task learning to classical single-task machine learning methodologies. Our analysis involved exploring the applicability of heterogeneous and sparse datasets of small-molecule kinase inhibitors to develop such models and how the various characteristics of these datasets impact model performance. The applicability and reusability of large public datasets are of considerable interest because of the wide diversity of screening technologies and data processing pipelines and available metadata annotations [31]. These datasets encompass a diverse array of factors, including assay methods, detection technologies, assay kits, reagents, experimental conditions, normalization, curve fitting, and more, as described in BioAssay Ontology [33]. It is worth noting that these details often vary considerably among the tens of thousands of protocols and thousands of laboratories that generated these datasets. In many cases, these details are not available in a standardized format and must be manually curated from publications or patents. 

We compare the performance of this MTDNN method against more traditional single-task machine learning approaches in the scope of virtual screening. While it is ideal to build models only from confirmed actives and inactives (KA–KI approach), published data extracted from journal publications and patents, which underlie our datasets, tend to exhibit a strong bias towards reporting active compounds, while inactive compounds are typically not considered as interesting. In practice, as evident, for example, in experimental high-throughput screening (HTS), even of focused libraries, most compounds are typically inactive. Therefore, the utilization of the KA–KI approach, while highly predictive in many contexts, is of limited practical relevance in the context of the virtual screening of very large datasets. While considering all data tested against unique protein kinases that do not have reported biological activity as presumed inactives (KA–PI approach) may introduce some errors, they are likely not overwhelming because active compounds are typically screened against the most similar likely off targets and the activity (including a lack thereof) is typically reported to characterize selectivity. Any errors introduced by the KA–PI assumption would be expected to degrade model performance. Therefore, the reported results could be considered conservative in the context of these assumptions.

Our results demonstrated a consistent and reliable predictive performance across most kinase tasks when a sufficient number of active compounds or structure–activity data points were available. The MTDNN exhibited a strong performance across all kinase groups, which considerably vary with respect to the available data. Kinases within the same group exhibit a higher similarity to one another by sequence compared to kinases across different groups [34]. The MTDNN demonstrates its ability to effectively leverage information from similar tasks, improving predictions for all tasks, including those that are underrepresented, by capturing shared knowledge and leveraging the similarities between closely related kinase groups. This is a capability that single-task methods cannot achieve. This characteristic of MTDNN is likely advantageous in the case of kinase-focused datasets, where the pronounced similarities of kinase ATP-binding sites and the well-established cross-activity of many small-molecule kinase inhibitors come into play. 

Similarity in molecular structure plays an important role in the performance of MTDNNs [35]. Our investigation highlighted that the MTDNN performs particularly well for kinase tasks that share similar compounds with other kinase sets. One of the key factors contributing to the superior performance of MTDNN could be its ability to adaptively learn molecular feature representations for active compounds across different classes, allowing for the model to better learn the distinct representation between all compounds. Specific reasons for such a performance remain elusive, emphasizing the need for further investigation to delve deeper into those observations. 

The primary goal of this study was to explore the applicability of this method in digging out small numbers of actual actives from a very large pool of inactives through virtual screening. This exploration involved leveraging diverse, target family-focused datasets of small-molecule kinase inhibitors. Importantly, our focus diverged from emphasizing the perfect classification of tested compounds into active and inactive classes. In this context, the enrichment factor (EF) stood out as a better choice to evaluate the model performance among various metrics available. The EF measures the ratio of the proportion of active compounds found within a very small fraction of predicted most likely actives in the library to the proportion of active compounds in the overall library. In other words, it quantifies how efficiently a model can enrich the active compounds at the top of the ranked list. When conducting virtual screening, the primary objective is to identify a small subset of compounds from a large library that is most likely to be active against a specific target. In this context, the EF provides a direct measurement of how well a model performs in prioritizing potentially active compounds. Enrichment is essentially a measurement of precision and recall for the initial small fraction considered. It is a useful measurement for highly imbalanced datasets and is particularly practical and established in virtual screening. The relative enrichment factor normalizes absolute enrichment based on the maximum possible enrichment, which depends on the ratio of active vs. inactive compounds and can be used to compare across different datasets. By considering unknown activity for any kinase as inactive, our approach can be considered practically relevant, as inactives vastly outnumber actives and many inactives are highly similar to actives; in some cases, presumed inactives may be active, leading to a degraded performance and more conservative results.

In this study, we utilized a pActivity threshold of 6 (corresponds to 1 μM) to distinguish between active and inactive inhibitors. The rationale behind selecting this specific cut-off value primarily originates from our comparison to previous high throughput screening hit-finding campaigns, where hits of greater than 1 μM are often deprioritized due to assay artifacts and solubility concerns. Upon analyzing the model performance across different cut-off values (pActivity of 7, 8, and 9) in conjunction with the threshold of 6 (Appendix A), we found that there is a modest improvement from cutoff 7 (100 nM). However, further improvement with cutoff 8 (10 nM) and cutoff 9 (1 nM) do not seem to yield a better performance. This may be attributed to the insufficient data for cutoffs 8 and 9 to demonstrate a better performance, whereas cutoff 7 may be close to optimal for minimizing the false negatives in the training data while still keeping a substantial number of actives. The optimal active classification threshold requires careful consideration, and future investigations are essential to understand the trade-offs and limitations associated with varying thresholds, considering both the model performance and practical implications. 

While the MTDNN performed well across the diverse small-molecule datasets investigated and appears applicable for virtual screening, limitations in predicting activities of highly dissimilar compounds, as observed in the IDG-DREAM Challenge datasets, have to be acknowledged. Specifically, the disparity in chemical structures between the training dataset and the IDG kinase tasks posed challenges to the model’s ability to make accurate predictions. This highlights the need for further studies and improvements to enhance the performance in scenarios where substantial differences exist in chemical structures between the training dataset and the target datasets. Future studies should explore strategies to address this limitation, such as incorporating additional training data that better represents the chemical space of the target datasets or employing transfer learning techniques to adapt the model to different chemical contexts.

It is also worth noting that, while the MTDNN approach yielded the best results and appeared most generalizable, it does come with considerably higher computational costs. However, with the latest generation of GPUs and as available datasets increase in size and diversity, deep neural networks are becoming increasingly relevant to fulfill the promise of virtual screening. Overall, our study contributes to the growing body of research in the field of virtual screening and underscores the potential of deep neural networks to further advance this area of drug discovery. Continued improvements and refinements in the application of these models will undoubtedly pave the way for more accurate and efficient virtual screening approaches in the future. 

## 4. Methods and Materials

### 4.1. Data Aggregation

Small-molecule activity data against the human kinome were obtained and curated from ChEMBL (release 21) and Kinase Knowledge Base (KKB) (release Q12016), which are large general and kinase-specific bioactivity data sources. 

To remove redundancy and inconsistency in the molecular representations, canonicalization or standardization is required to generate a unique SMILES representation for each molecule. An in-house chemical structure standardization protocol was implemented using Biovia Pipeline Pilot (version 20.1.0.2208). Salts/addends and duplicate fragments were removed so that each structure consisted of only one fragment. Stereochemistry and charges were standardized, acids were protonated and bases deprotonated, and tautomers were canonicalized. Stereochemistry and E/Z geometric configurations were removed to compute extended connectivity fingerprints of length 4 (ECFP4) descriptors (see Section 4.3) for compounds since these are not differentiated by standard ECFP fingerprints. 

After small molecules were preprocessed and normalized, UniProt identifiers were obtained by mapping domain information to UniProt using the Drug Target Ontology (DTO), resulting in 485 unique UniProt IDs. All UniProt IDs were then mapped to corresponding ChEMBL and KKB target IDs. Note that mutants were excluded from the ChEMBL and KKB datasets based on the existence of their variant labels. The bioactivity annotations obtained from ChEMBL were filtered by assay annotation confidence score ≥ 5, and only compounds with activity annotations corresponding to a standard type of Kd, Ki, IC50, or Potency were accepted. The resulting data were then aggregated by chembl_id, standard_type, and UniProt_id. The median standard_value was calculated and then transformed using a –log_10_-transformation. 

Commercial kinase bioactivity data from KKB were also obtained, filtered, and aggregated by unique compounds, endpoints, and targets. The overlapping compound, target annotations, and endpoints in ChEMBL and KKB data were aggregated by their median. Compounds were then grouped by canonical SMILES and received an active label for each kinase where an aggregated pActivity value ≥ 6 was observed and received no label otherwise. Kinase labels with <15 unique active compounds were removed, leaving 668,920 measurements (315,604 unique compounds) distributed across 342 unique kinase classes for which to build models. The compounds utilized in this study predominantly represented pre-clinical compounds (>99%), featuring comprehensive bioactivity data sourced from the literature and patents (Appendix A). The aggregated dataset distribution was slightly skewed towards inactive compounds; 47% of the 668,920 training examples were active (about 315 K). The specific data curation procedure and target and compound overlap before removing kinase labels with <15 unique active compounds are shown in Figure 1.

### 4.2. Dataset Processing

The rationale behind gathering targets from public and commercial sources was to amass a collection of data that could leverage the power of both deep learning and multi-task algorithms. Given a set of related tasks, a multi-task network has the potential for higher accuracy as the learner detects latent patterns in the data across all tasks. This makes overfitting less likely and makes features accessible that may not be learnable by a single task. 

We built the kinase predictors based on two different methods of handling kinase datasets. One method trained single- and multi-task classifiers from known actives and known inactives (KA–KI), which utilized only the reported active and inactive compounds for each task. The other method built single- and multi-task classifiers using all available data for unique kinase molecules, termed known active and presumed inactives (KA–PI). The KA–PI models were built by identifying for each kinase task the active compounds and treating the remaining compounds as inactive (decoys). For the purpose of this study, KA–PI was used for model building and evaluation for the KA–PI model, and KA–KI was trained for the KA–KI model but was evaluated on the larger KA–PI set. Both methods utilized all datasets and were characterized using the metrics introduced below. 

### 4.3. Small-Molecule Topological Fingerprints (Features)

Extended connectivity fingerprints (ECFP4) were calculated using RDKit toolkits in Python. The ECFP4 algorithm assigns numeric identifiers to each atom, but not all numbers are reported. The fingerprints were hashed to a bit length of 1024; therefore, very similar molecules can both be assigned the same numeric identifiers. Although increasing the number of bits reported can reduce the chances of a collision, there are also diminishing returns in the accuracy gains obtainable with longer fingerprints (e.g., 1024-bit, 2048-bit, or larger fingerprints can be used). This and computational complexity concerns were the pragmatic reasons why we chose to use 1024-bit ECFP4 fingerprints.

### 4.4. Cross-Validation Approach

To evaluate the predictive performance of the multi-task models, we implemented three different 5-fold cross-validation strategies, including splitting by scaffold, by molecular weight, and randomly. In virtual screening, it is important to consider the chemical diversity of the training and test sets for domain applicability and for evaluating how well the classifier generalizes to new chemotypes. For the scaffold-based cross-validation, we performed hierarchical clustering for all compounds using Biovia Pipeline Pilot (version 20.1.0.2208), specifying approximately 300 clusters with an average of approximately 1000 compounds. The pairwise Tanimoto similarities were calculated between all cluster centers and visualized to ensure that chemical dissimilarity was sufficient (Appendix A). Each cross-validation held out 1/5 of the scaffolds, and mean performance was calculated. Molecular weight was another distinguishing feature of compounds that could be used to estimate classifier performance. This method aims to keep the classifiers from overfitting on compound size, and molecular weight can also be considered a simple surrogate for how different compounds are. Molecular weight was calculated in Python using RDKit. Compounds were sorted by increasing molecular weight, and 1/5 of the dataset was held out during each training iteration. Randomized 5-fold cross-validation was also performed using a random stratified splitter to split the ChEMBL and KKB aggregated data into train, valid, and test sets with 80%, 10%, and 10% accordingly for the evaluation of the multi-task model performance. By running random stratified cross-validation within one dataset, all 3 sets share common compounds and kinase targets. 

### 4.5. Model Construction

#### 4.5.1. Multi-Task Artificial Neural Network Architecture

Neural networks can produce impressive non-linear models for classification, regression, or dimensionality reduction and are applicable in both supervised and unsupervised learning situations. Neural networks take as input numerical vectors and render input to output vectors with repeated linear and non-linear transformations experienced repeatedly at simpler components called layers. Internal layers project distributed representations of input features that are useful for specific tasks.

More specifically, a multiple hidden layer neural network is a vector-valued function of input vectors *x*, parameterized by weight matrices *W_i_* and bias vectors *b_i_*:$$x_{0}=x\;z_{i}=W_{i}x_{i-1}+b_{i}\;x_{i}=f_{i}\left(z_{i}\right)$$
where *f_i_* is nonlinear activation function, such as rectified linear unit (ReLU) (max[0, *z_i_*]), *x_i_* is the activation of layer *i*, and *z_i_*is the net input to layer *i*. After traversing *L* layers, the final layer *x_L_* is output to a simple linear classifier (in our case, the softmax) that provides the probability that the input vector *x* has label *t*:$$P\left(y=t\right|\;x)=\frac{e^{{\left(w_{t}\right)}^{T}(x_{L})}}{\sum_{m=1}^{M}e^{{\left(w_{m}\right)}^{T}(x_{L})}}$$
where *M* is the number of possible labels in our case of binary prediction for each task (*M* = 2) and *w_t_*, *w_m_* are weight vectors; *w^T^x* is the scalar (dot) product. Our network therefore takes a numerical chemical fingerprint descriptor of size 1024 as input, one or multiple layers of ReLU hidden units, and softmax output neurons for each kinase class or task. Given the known input and output of our training dataset, we optimized network parameters (*x*, *y*) = ({*W_i_*}, {*b_i_*}) to minimize a defined cost function. For our classification problem, we used the cross-entropy error function for each task:$$C\left(x_{L},\;y\right)=-\sum_{t}^{T}w_{t}(y_{t}\log\left(f_{t}\left(x_{L}\right)\right)+\left(1-y_{t}\right)\log(1-f_{t}\left(x_{L}\right)))$$
where *T* is the total number of tasks, kinase classes in our implementation. The training objective was therefore the weighted sum of the cross-entropies over all kinase targets.

The algorithm was implemented in Python using the Keras package with Theano backend and was run on Nvidia GeForce GTX 1650 GPUs with 32GB RAM to increase performance. Hyperparameter optimization included adjustments of momentum, batch size, learning rate, decay, number of hidden layers, number of hidden units, dropout rate, and optimization strategy. The best-performing model consisted of training a batch size of 128 with two hidden layers of size 2000 × 500 using a dropout rate of 25% and a learning rate of 0.0003 across each hidden layer for stochastic gradient descent learning. These initially tuned hyperparameters were consistently applied throughout the study, revealing a commendable level of stability and robustness across various data split strategies. Model training varied in time from 1 day for all kinase classes to 2 h for the 100 kinases with the most active compounds.

#### 4.5.2. Single-Task Deep Neural Networks

A single-task deep neural network (STDNN) model architecture is designed to address a specific task or singular objective. The key distinction lies in the number of output tasks. An MTDNN has multiple output tasks, each consisting of two neurons, representing whether a compound is active against the target or not, whereas an STDNN is characterized by a single output task, emphasizing its focus on individual targets. The same set of hyperparameters used for MTDNN was applied for STDNNs. All STDNNs were executed on the Pegasus supercomputer at the University of Miami (https://idsc.miami.edu/pegasus/).

#### 4.5.3. Other Single-Task Methods

All single-task methods were implemented in Python using the Sci-kit Learn machine learning library. Methods included logistic regression, random forests, and Bernoulli naïve Bayes binary classifiers. Each method was implemented on the Pegasus supercomputer using stratified 5-fold cross-validation strategies and both KA–KI and KA–PI datasets but training each class individually.

#### 4.5.4. Multi-Task Non-DL Approach

A multi-task random forest was performed in Python as the multi-task non-DL method to compare the performance with the MTDNN model. The aggregated ChEMBL and KKB data were used to train the model, and the randomized cross-validation strategy was performed to split the dataset randomly into 5 folds, which would perform the fitting procedure five times in total, with each fit being performed on 90% as the train set and remaining 10% as the test set. Random forest fitted a total number of 100 decision tree classifiers on the train set. Two-sample t-test was used to compare the model performance between this random forest model with MTDNN based on their ROC score. 

### 4.6. Metrics and Model Evaluation

To adequately evaluate the machine learning models, we used a variety of metrics. The commonly used receiver operating characteristic (ROC) classification metric is defined as the true positive rate (TPR) as a function of the false positive rate (FPR). The TPR is the number of actives at a given rank position divided by the total number of actives, and the FPR is the number of inactives at a given rank position divided by the number of inactives for a given class. The area under the curve (AUC) was calculated from the ROC curve and is the metric we report. The calculation from a set of ranked molecules is given as follows:$$AUC=\frac{1}{n(N-n)}\sum_{i=2}^{N}A_{i}(I_{i}-I_{i-1})$$
where *n* is the number of active compounds, *N* is the total number of compounds, *A* is the cumulative count of actives at rank position *i*, and *I* is the cumulative count of inactives at rank position *i*.

Metrics including accuracy, precision, recall, and F1-score were also utilized to evaluate the model performance. The imbalanced datasets pose a significant challenge for binary classification models, as they can result in biased predictions towards the majority class. Accuracy measures the proportion of correctly classified samples over the total number of samples. However, accuracy can be misleading in the case of highly imbalanced datasets, as it can result in high scores even when the model fails to correctly classify minority class samples. Precision, recall, and F1-score are more suitable metrics for imbalanced datasets. F1-score considers both precision and recall in its calculation, where precision measures how accurate the model is in predicting the positive class, and recall measures how well the model identifies the positive class among all positive samples. The formulas for the specific evaluation indicators are as follows:$$Accuracy=\frac{{True}_{Positive}+{True}_{Negative}}{{True}_{Positive}+{False}_{Positive}{+True}_{Negative}+{False}_{Negative}}\;Precision=\frac{{True}_{Positive}}{{True}_{Positive}+\;{False}_{Positive}}\;Recall=\frac{{True}_{Positive}}{{True}_{Positive}+\;{False}_{Negative}}\;F1\;Score=\frac{2*Precision*Recall}{Precision+Recall}$$

Although precision, recall, and F1-score are suitable for imbalanced data, for the purpose of this study, in which we focused on virtual screening, the enrichment factor (EF) was a more preferred and popular metric used in the evaluation of virtual screening performance. EF denotes the quotient of true actives among a subset of predicted actives and the overall fraction of actives and can be calculated as follows:$$EF\left(X\right)=\frac{\sum_{i=1}^{N}\delta \left(r_{i}\right)}{nX},\;with\;\delta \left(r_{i}\right)=\left\{\begin{array}{c} 1,\;r_{i}\;\leq \;XN\\ 0,\;r_{i}>XN \end{array}\right.$$
where *r_i_* indicates the rank of the *i*th active, n is the number of actives, *N* is the number of total compounds, and *X* is the ratio within which EF is calculated. We evaluated the EF at 0.1% (*X* = 0.001) and 0.5% (*X* = 0.005) of the ranked test set. To evaluate the maximum achievable enrichment factor (EFmax), the maximum number of actives among the percentage of tested compounds was divided by the fraction of actives in the entire dataset. To more consistently quantify enrichment at 0.1% of all compounds tested, we report the ratio of EF/EFmax (Figure 5). This normalized metric is useful because EF values are not directly comparable across different datasets due to the maximum possible EF being constrained by the ratio of total to active compounds and the evaluated fraction as shown in the equation above.

### 4.7. Kinase Task Similarities

To evaluate how similar chemical structures in one kinase task are compared to those in all other kinase tasks, 25 diverse active compounds in each kinase task were selected using the diverse molecule component in Pipeline Pilot (version 20.1.0.2208). This algorithm defines diverse molecules by maximum dissimilarity using the Tanimoto distance function and ECFP4 descriptors. For each kinase task, the average maximum Tanimoto similarity to all other kinase tasks was calculated based on the 25 diverse samples from each class. This task-based similarity refers to the average maximum similarity of the reference class to all other kinase tasks as shown in Figure 8.

### 4.8. KINOMEscan and IDG Predictions

KINOMEscan datasets were obtained from the LINCS Data Portal (http://lincsportal.ccs.miami.edu/). The initial datasets contained 85 compounds with defined/known chemical structures and 387 different kinase targets. Kinase activity was screened at 10 µM compound concentration. Kinase domain targets were curated and standardized using the Drug Target Ontology (DTO, http://drugtargetontology.org/), and datasets were joined into a data matrix of unique kinase targets (domains) and standardized small-molecule canonical smiles. Of the total 387 kinase targets, a subset of 291 targets shared commonality with those included in the training data, which was utilized to evaluate the model’s performance; null values were introduced where no data was available. Kinase activity values were binarized for each small-molecule kinase inhibition value [0, 1], where 1 indicated active (≥80% inhibition) and 0 indicated inactive. The resulting dataset was imbalanced with ~83% inactives and ~17% actives. Accuracy was calculated as described above.

IDG-DREAM round 1 and 2 pKd kinase inhibitor datasets were collected from Drug Target Commons (https://drugtargetcommons.fimm.fi/). After mapping kinase domain information to the UniProt identifiers via DTO, a total number of 808 data points was obtained for 94 compounds with 288 unique kinase targets in the dataset. From these, 223 kinase targets, which aligned with the targets present in the training data, were kept for testing purposes. An amount of ~63% of data points were labeled as actives using a cut-off of pKd (−logM) of 6, where the actives were defined as data points with pKd greater than or equal to 6 and inactives (~37%) were those with pKd less than 6. ROC scores were calculated as described above. For both KINOMEscan and IDG chemical structures, the same ECFP fingerprint descriptor of size 1024 was used.

## Figures and Tables

**Figure 1 ijms-25-02538-f001:**
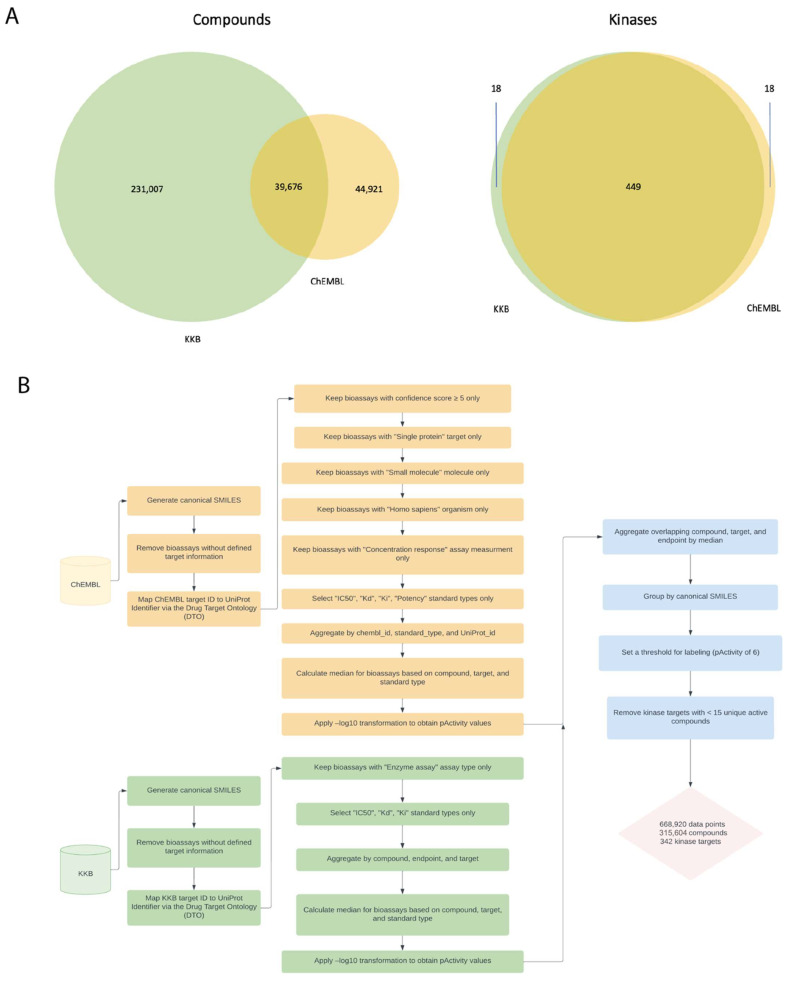
Data aggregation workflow and overlap between test datasets. (**A**) Overlap between ChEMBL and KKB unique kinase inhibitors and targets. (**B**) ChEMBL and KKB small-molecule bioactivity data aggregation workflow.

**Figure 2 ijms-25-02538-f002:**
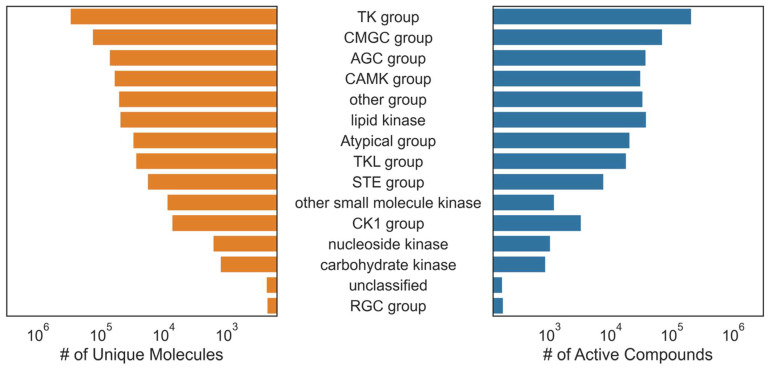
Distribution of dataset compounds by kinase family. The bar plot shows the sum of all molecules with kinase bioactivity annotations on the left. The sum of all active compounds for each kinase is shown on the right for each kinase category.

**Figure 3 ijms-25-02538-f003:**
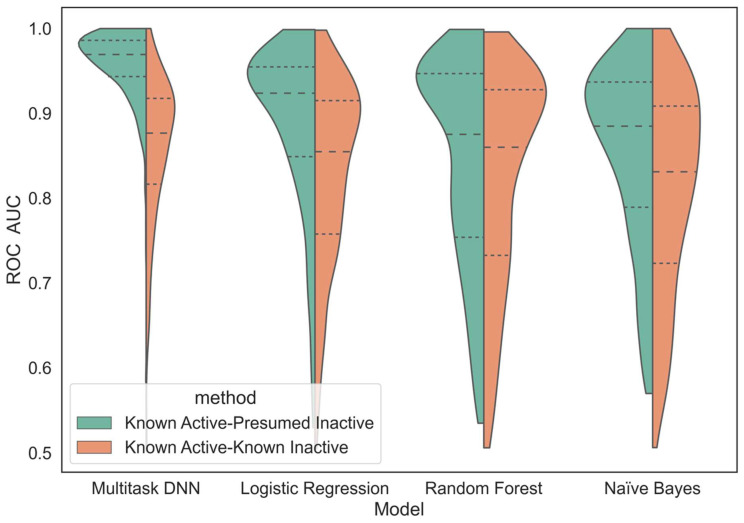
Differences in ROC score for kinase learning methods. Split violin plots with ROC score data distribution for all 342 kinase tasks. Each split violin plot represents a different modeling method with Known Active–Presumed Inactive (KA–PI) and Known Active–Known Inactive (KA–KI) datasets shown on the left and right, respectively. The receiver operating characteristic ROC area under the curve (AUC) is a metric to evaluate how well the model predicts active compounds (true positives) vs the rate of incorrectly predicted actives (false positives). A higher score dictates a better discrimination of the model between true and false predictions.

**Figure 4 ijms-25-02538-f004:**
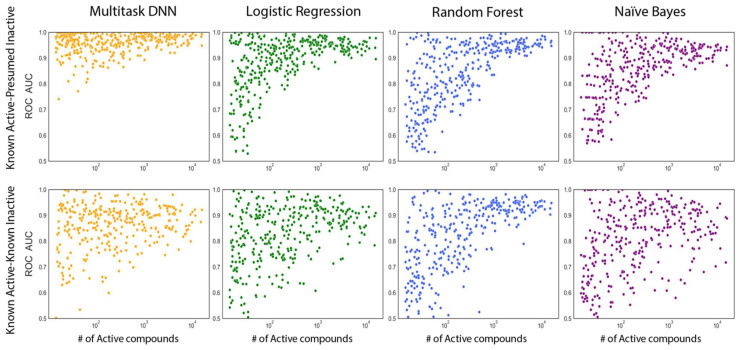
Effect of the number of active compounds model performance. Scatter plots ROC score data distribution for all 342 kinase tasks. Each box represents a specific model using either the Known Active–Known Inactive (KA–KI) or Known Active–Presumed Inactive (KA–PI) datasets.

**Figure 5 ijms-25-02538-f005:**
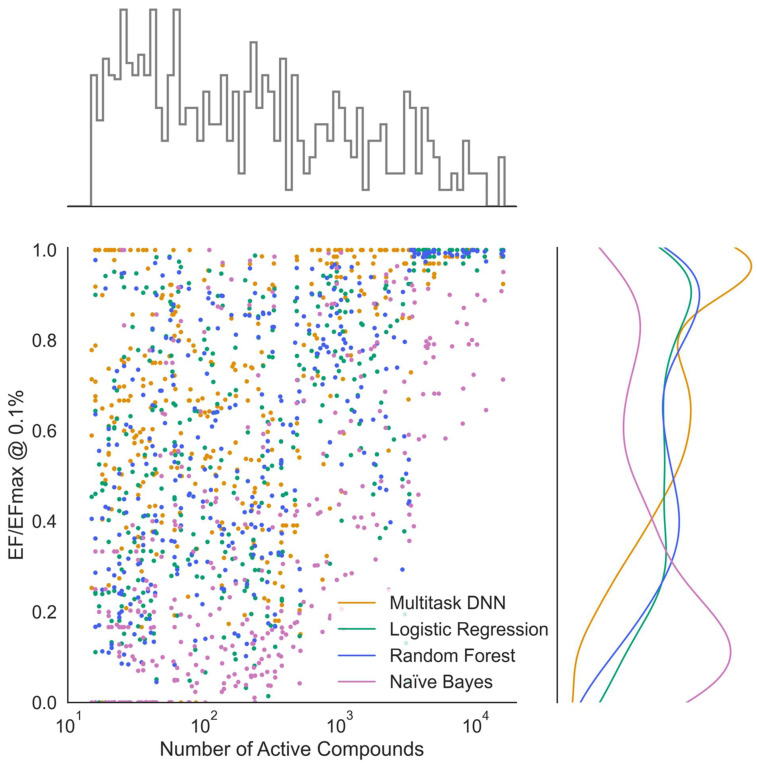
Normalized enrichment factor with respect to the number of active compounds. Known Active–Presumed Inactive (KA–PI) model performance across machine learning methods measured by normalized enrichment factor (EF/EFmax) as a function of the number of active compounds for each kinase task.

**Figure 6 ijms-25-02538-f006:**
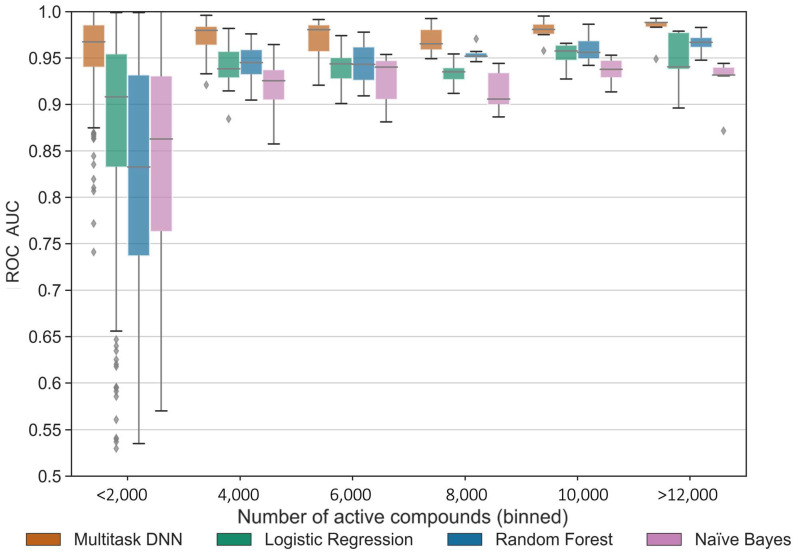
The effect of the number of active compounds on performance across learning methods. ROC score for different machine learning methods binned by ranges of active compounds across 342 kinase tasks (ROC score averaged over 5 repetitions of 5-fold random cross-validation).

**Figure 7 ijms-25-02538-f007:**
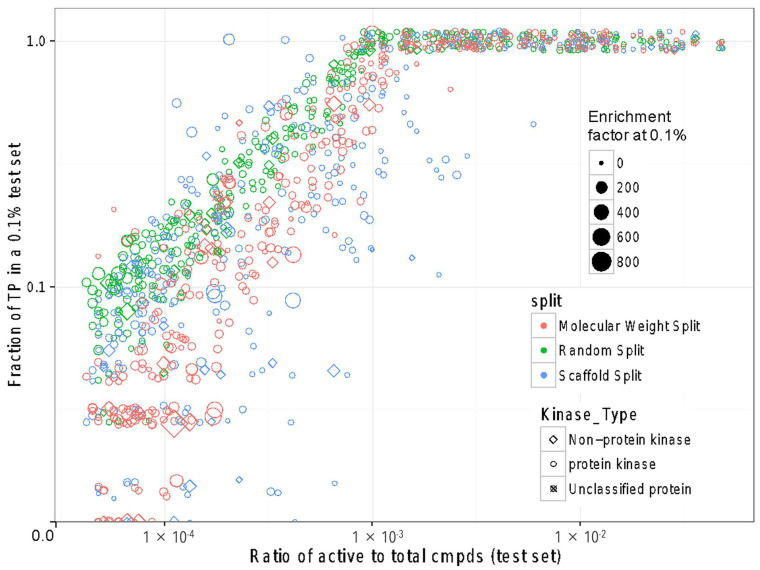
The fraction of recovered true positives among 0.1% tested compounds depending on the fraction of actives in the total dataset. Scatterplot of the fraction of true positives at 0.1% of the test set for all 342 Known Active–Presumed Inactive (KA–PI) MTDNN kinase tasks across different cross-validation splitting strategies shown in different colors. Each point represents a specific kinase task. The size indicates an enrichment factor at 0.1%. The shapes correspond to non-protein, protein, and unclassified kinases. The axes are log–log scale.

**Figure 8 ijms-25-02538-f008:**
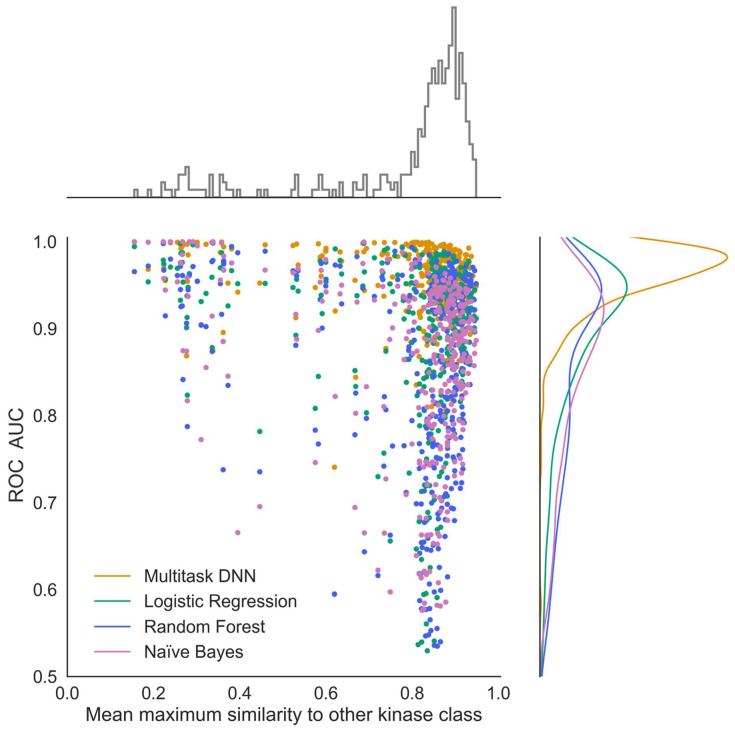
Effect of dataset similarities on model performance for multi-task vs. single-task methods. Known Active–Presumed Inactive (KA–PI) model performance across machine learning methods measured by ROC score as a function of the average maximum similarity of compounds of one kinase task compared to all other kinase classes.

**Table 1 ijms-25-02538-t001:** Statistical Significance Analysis of Model Performance using ROC Scores.

Machine Learning Methods	*p*-Value vs. KA–PI MTDNN	Average ROC
KA–PI LR	<2.2 × 10^−16^	0.887
KA–PI RF	<2.2 × 10^−16^	0.842
KA–PI NB	<2.2 × 10^−16^	0.852
KA–PI MTDNN		0.958

Abbreviations: LR = logistic regression; RF = random forest; NB = naïve Bayes; MTDNN = multi-task deep neural networks.

## Data Availability

ChEMBL (https://www.ebi.ac.uk/chembl/) and KKB (www.kinasedb.com) aggregated dataset is available in the Raw data folder on GitHub. The IDG Challenge Round 1 and 2 pKd dataset obtained from DrugTargetCommons (https://drugtargetcommons.fimm.fi/) and the KINOMEscan dataset obtained from LINCS Data Portal (http://lincsportal.ccs.miami.edu/) are available in the same Raw data folder. The codes for reproducing the results are available on GitHub (https://github.com/jxh10111/Kinome-wide-Virtual-Screening-by-Multi-task-Deep-Learning). Key packages used and their versions are listed in the GitHub repository.

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
