# Peer review of "Kinome-Wide Virtual Screening by Multi-Task Deep Learning"

_ijms, 2024, doi:10.3390/ijms25052538_

Round 1
Reviewer 1 Report
Comments and Suggestions for Authors
Author Response
Dear Reviewer,
Thank you very much for taking the time to review this manuscript. Please find the detailed responses in the attachment.
Sincerely,
Stephan

Reviewer 2 Report
Comments and Suggestions for Authors
The authors in their study have presented an important study that can make the implementation of polypharmacology realistic and can be adapted to other protein classes. The research questions were addressed and answered.
However, my concern is that while the workability and accuracy of the model were proven the conclusion was not too specific to a point where they pinpointed a particular or set of compounds (name, structure, chemical ID, or SMILEs) that could potentially exhibit polypharmacology activity against the target kinases. These could further be experimentally validated in further studies for their polypharmacological potencies.
Also, can the authors clarify if the compound libraries cut across different stages of drug development (pre-clinical to clinical) or are mainly FDA-approved? Do they contain discontinued compounds based on toxic bioactivity that will be subject to further optimization?
Author Response

(The authors gave the same response as above.)
